# The Social, Behavioral, and Ethical Modalities of COVID-19 on HIV Care in South Africa: A Systematic Review

**DOI:** 10.3390/ijerph19159766

**Published:** 2022-08-08

**Authors:** Khushali Roy, Aliayah Himelfarb, Kapil Karrah, Laura Porterfield, Lauren Paremoer, Hani Serag, Wei-Chen Lee

**Affiliations:** 1School of Medicine, University of Texas Medical Branch, Galveston, TX 77555, USA; 2Department of Psychiatry, Weill Cornell Medicine, New York, NY 10021, USA; 3Medical College, MedCiti Medical Institute, Ghanpur 501401, Telangana, India; 4Department of Family Medicine, University of Texas Medical Branch, Galveston, TX 77555, USA; 5Department of Political Studies, University of Cape Town, Cape Town 7700, South Africa; 6Department of Internal Medicine, University of Texas Medical Branch, Galveston, TX 77555, USA

**Keywords:** HIV, COVID-19, district health information system, service delivery, stigma, lockdown, ART disruptions, clinic visits

## Abstract

The profound public health impact of the novel outbreak of the SARS-CoV-2 virus in 2019 has been unparalleled in the last century. Rapid spread of the disease and a high death toll fueled the development and global rollout of effective vaccines regardless of the massive inequitable access. However, some public health measures intended to control COVID-19 have had collateral effects on the control of other infectious diseases. In this systematic review, we analyze the impact of the COVID-19 pandemic on efforts to control HIV in South Africa, emphasizing the social, ethical, and behavioral ramifications. The SCOPUS, PubMed, Ovid, PsychINFO, and Cochrane Library databases were searched for publications between March 2020 and January 2022. Of the 854 articles identified, 245 were found duplicated, and 609 were screened, 241 of which were potentially eligible, and 15 of which were ultimately included. Although no studies on the ethical implications were eligible for our study criteria due to insufficient primary data to perform an analysis on, we explored this topic in the Discussion section of this paper. We confirm declines in ART, PrEP, and HIV testing during the initial lockdown period, with slight variations across the South African provinces. Protecting routine services and reducing the disease burden on high-risk nations such as South Africa is imperative moving forward with the pandemic.

## 1. Introduction

### 1.1. COVID-19 and HIV

The COVID-19 pandemic continues to take a toll on public health around the world, with many lower- and middle-income countries continuing to experience an acute need for resources and interventions. At the onset of the pandemic, many countries struggled to plan and mobilize task forces to control the spread of disease, encountering several pitfalls, including substandard administration and lack of ownership of responsibility, as well as inefficient resource management on many levels [1]. Pre-pandemic deficiencies in healthcare systems and social welfare programs were amplified, especially in areas already affected by preexisting public health crises. Additionally, the pandemic revealed several drawbacks of for-profit healthcare systems, which tend to rely on a stable cash flow from elective procedures and non-acute care [2]. This higher-risk model of business is particularly vulnerable during lockdown periods, which prevent returns of equity and worsen financial sustainability. 

Analyzing infectious disease management past the workings of the current pandemic is imperative. As the long-standing epicenter of the HIV epidemic, South Africa has led major advancements in HIV research, including the development of the largest antiretroviral treatment program in the world [3]. This program has been a source of national pride, as South Africans approached the UNAIDS 90–90–90 targets (90% tested, 90% on treatment, and 90% suppressed virally) at 85–71–86, estimated by the South African National HIV Prevalence, Incidence, Behavior and Communication Survey [4]. However, even amidst this progress, an additional 200,000 new HIV infections and 72,000 AIDS-related deaths were reported in South Africa in 2019 [5].

### 1.2. Historical Context of HIV in South Africa

Only a few decades ago, historical leaders in South Africa denied the existence of AIDS and fought against providing antiretroviral therapy (ART) for infected individuals [6]. In 1994, concerns of HIV in South Africa were overshadowed by the transition from apartheid [3]. In this time, the prevalence of HIV grew extensively, along with its denialism by President Thabo Mbeki and Minister of Health Manto Tshabalala-Msimang [6,7]. The introduction of the prevention of mother-to-child transmission (PMTCT) was one of the first prophylactic strategies that was deemed both affordable and effective, but Pres. Mbeki and his Health Minister blocked universal access to PMTCT until a Constitutional Court decision forced them to change this policy [3]. In 2002, ART programs came into the spotlight when the Cabinet of South Africa opposed Mbeki’s denialist stance, releasing a statement recognizing the link between HIV and AIDS [3,7]. Delays in making ART universally accessible were significantly attributed to the international patent regime contained in the Trade-Related Aspects of Intellectual Property Rights (TRIPS) framework of the World Trade Organization [8]. It initially prevented the production of generic medicines before the 20-year protection of brand medicines. The organized movement of people living with HIV and the support of public interest in civil society, e.g., the Treatment Action Campaign, have contributed massively to this change, obligating the WTO member states to approve the TRIP’s flexibility in the Doha round [9]. In this round, the TRIP’s flexibility agreed on compulsory license and parallel importation, essentially changing the game. In addition, the behavior of donors, specifically the refusal of the use of their funds to purchase generic medicine from other countries in the Global South (e.g., India), also impeded universal access to ART [10]. For example, the U.S. President’s Emergency Plan for AIDS Relief (PEPFAR) used only 5% of its allocated budget for drugs to buy generic medicines in 2004 and 2005 [11]. This was simply by instituting a rule to restrict the use of funding to the purchase of ART approved by the U.S. Food and Drug Administration. 

Since early denialism, ART programs, pre/postexposure prophylaxis research, and increasing contraceptive access have significantly contributed to combating the HIV epidemic [4]. However, despite the progress made in recent decades, the rate and distribution of HIV infection in the South African population persists in a heavily disproportionate manner compared to the rest of the world, with factors ranging from socioeconomic status to HIV stigma shaping access to care [7].

### 1.3. Defining Social, Behavioral, and Ethical Implications

When we look historically at the extent and patterns of disease in Western society, we see a dramatic shift from infectious diseases to noncommunicable diseases [12]. As is evidenced by declining immunization rates in high-income countries, the fear of death and disability resulting from infectious diseases has declined, even as new communicable conditions such as AIDS and COVID-19 have emerged and spread rapidly and globally. Rather than focusing on individual experiences, the study was designed to understand how COVID-19 has impacted efforts to manage the HIV epidemic through social, behavioral, and ethical perspectives. The social meanings of an illness can reflect a nation’s policy, economy, culture, class, and social norms [8]. These affect a large number of people, and the individuals in society may not have control over social issues. Ethics is about the moral conduct or moral principle that may determine what is good and bad or right and wrong [13]. Clinical ethics refers to an application of science and understanding of morality in the fields of medicine and health sciences [14]. Unethical clinical research may affect the scientific validity and merit of the research. Health behaviors refer to actions taken by individuals that affect health or morality [15]. It may also refer to the behaviors of healthcare providers that contribute to a patient’s outcomes [16]. These ideas can be viewed as a graph on Figure 2. In addition to three distinct areas, our study recognized the interactions among social, ethical, and behavioral issues, such as how social environments shape health behaviors or how ethical concerns affect social norms. By systematically reviewing the literature in the social, ethical, and behavioral fields, this study will address the impacts of the COVID-19 pandemic on the HIV epidemic. Furthermore, we aim to provide an argument that promotes and facilitates better responses to managing HIV/AIDS services during health emergencies.

## 2. Materials and Methods

### 2.1. Search Strategy 

Following the guidelines of the Systematic Reviews and Meta-Analyses (PRISMA) 2020 diagram, we searched the SCOPUS, PubMed, Ovid, and Cochrane Library databases between March 2020 and January 2022 for studies published in English (Figure 1, page 11). Search strings were exported to Endnote, and duplicates were identified before screening. Several combinations of search terms were used, including subject heading terms/keywords relevant to COVID-19 (e.g., SARS-CoV-2 OR Coronavirus Disease 2019 OR COVID-19 OR severe acute respiratory syndrome coronavirus 2 OR coronavirus infection) and HIV (e.g., HIV OR human immunodeficiency virus) (see full search strategy here: https://tinyurl.com/43wm6478, accessed on 22 June 2022).

### 2.2. Inclusion and Exclusion Criteria

Studies in the English language, RCTs, meta-analyses, systematic reviews, observational studies, and articles published in journals with low or no impact factors were considered. We included articles in full text and peer-reviewed papers. Studies looking at the pandemic interplay with the tuberculosis epidemic were considered due to interrelations between both pandemics. Other included studies discussed the disruptions to treatment programs, studies with prospective analyses on HIV and COVID in the near future, and how the goals of ending the HIV epidemic were affected by the onset of COVID-19. Articles only discussing the comorbidity risks between COVID and HIV without reference to the social, behavioral, and ethical implications or lack of specificity in the effects on South Africa were not included. Furthermore, studies with a lack of primary literature were not considered. 

### 2.3. Data Collection

All search strings of the databases were exported to EndNote for citation management. In the searching of the four databases, we retrieved 854 references, 435 of which were duplicates. Upon deduplication of the data in the Endnote software, 245 studies were removed. This left 609 studies to screen from for the systematic review. Three hundred and sixty-eight records were excluded in EndNote, a process that was done by filtering the exported results to include “coronavirus”, “COVID-19”, or “SARS-CoV-2” as keywords in the full-text paper. This left 241 studies eligible; after which, 12 were excluded due to the information being limited to comorbidities of HIV and COVID-19. Lastly, 229 commentary papers and articles were excluded that did not have primary data, addressed results only pertinent to COVID-19 or failed to address South Africa specifically. Four studies were included that were hand searched using the Google search engine. This concluded with a total count of 15 papers that were assessed and included for analysis in our review.

### 2.4. Topic Modeling

During the search of the literature, a plethora of information was found on delays and interruptions in treatments in relation to COVID-19. Filters were applied to include the timeline of the pandemic to keep the relevance of the effects of COVID-19. Due to the broad nature of the search, we performed the search in accordance with the social, behavioral, and ethical terms of illness, which were defined in the Introduction section of this paper. Two people extracted and checked the data, with both reviewers independently extracting the data. A full list of the keywords is attached to the link on the search strategy.

### 2.5. Quality Assessment

Data synthesis was measured with a quality assessment and risk of bias tool. We used the Critical Appraisal Skills Program, which provides a thorough checklist to assess the quality, validity, and biases of each study according to the study design. We completed the CASP tool checklist for cohort studies (12), listed in Table 1, systematic reviews (2), listed in Table 2, and qualitative studies (1), listed in Table 3. Furthermore, an analytical breakdown of the studies is provided in Table 4.

## 3. Results

In our systematic review of the literature, four articles were found on disruptions in antiretroviral therapy. Several of these studies supported data on increased morbidity with ART disruptions, emphasizing the necessity for alternative delivery methods of therapy. Three papers on alternative service delivery discussed the accelerated response of differentiated HIV services, fees of the home delivery of medications, and the involvement of community models and community healthcare workers to increase the comprehensive treatment access. Five articles described the disruptions in HIV testing in regular primary care visits and antenatal visits. One article discussed the availability and demographics of contraceptive access during the pandemic, as well as culture and stigma. Lastly, one article expanded on PrEP disruptions and one reviewed symptomatic admission. Several articles overlapped in these topics, which are explored appropriately in the Discussion section of this paper. 

## 4. Discussion

Due to the vast amount of literature intertwining the social and behavioral aspects of the HIV response, we analyzed these studies through the combined socio-behavioral lens. Ultimately, although two studies discussing ethical modalities of our topic were found, there was no primary data to be considered eligible in our review. A review of terms defining the social, behavioral, and ethical lenses can be viewed in Figure 2.

### 4.1. Disruptions in Care

#### 4.1.1. Antiretroviral Therapy (ART)

For people living with HIV (PLHIV), managing the disease requires a lifelong administration of ART. Although ART can be effective, if a consistent adherence is not maintained, viral resistance can be a significant concern. The drug toxicities and side effects of ART have caused myriad adverse events (AE). Several studies have shown that unplanned interruptions of ART can increase the risk of such AEs, as patients exhibit lower CD4 counts and compromised immunological defenses [30]. One data set extrapolating an analysis of nearly 60% of the public sector facilities in South Africa estimated 310,000 fewer initiations of ART in 2020 compared to the previous year [31]. These implications point to the increase in the number of patients who later initiate ART at more advanced stages of the disease and with lower CD4 counts, jeopardizing their prognosis in ART [17]. Furthermore, South Africans with comorbid tuberculosis and HIV experienced persistent immune dysfunction when infected with COVID-19, even when previously treated with ART [18,32]. Thus, ART maintenance is critical to management of the disease for PLHIV. 

During the COVID-19 pandemic, many provinces of South Africa documented changes in the course of ART treatment. A convenient sample of people receiving HIV treatment services in Cape Town, South Africa recorded that more than one in three participants experienced interruptions in their HIV care, including barriers to access to medications, a lack of service delivery, failure to attend a clinic, and the shifting of clinic services away from HIV and towards COVID treatment in the Western Cape [19]. On the other hand, this study found a cross-sectional correlation between practicing COVID-19 preventative behaviors and having a higher adherence to ART. 

In the Province of KwaZulu-Natal, a significant drop in clinic visits during lockdown was immediately followed by a 20% increase in visits for HIV over the original baseline. The higher utilization may reflect the increase in demand due to the interruption of services during the lockdown [3,4]. Overall, rural clinics were less impacted by the lockdown and had better success in maintaining HIV services for patients already receiving ART [1]. Clinics in urban areas had greater struggles in initiating ART during this period. As a result of these ART disruptions, an increase in HIV visits created an imbalance in the supply and demand in anticipation of interruptions to HIV medication availability [33].

#### 4.1.2. HIV Testing

Common measures of transmission prevention include in-person facility HIV testing, community testing, and at-home testing coupled with linkage to care. While at-home testing is available in South Africa, it is still not commonly used. Over the years, the testing availability and access have greatly improved, although gaps continue in lower socioeconomic areas. One of the most pressing issues affecting transmission is the number of patients who go undiagnosed every year. This count of “missing patients” is significantly increased with the disturbances in HIV testing, playing a critical role in the spread of the disease [17]. 

A study on primary care visits in several South African provinces revealed a 22.3% decline in HIV testing numbers in 2020 [34]. Decreased testing was also correlated to decreases in ART initiation, setting back goals by the National Strategic Plan of South Africa to end the epidemic within five years. During the pandemic, inter-district variations in neonatal HIV PCR testing were found in KwaZulu-Natal, suggesting continued inequalities in maternal care [21]. Limpopo Province saw contrasting data, as PCR indicators from the District Health Information Systems showed no changes during lockdown. This database revealed that the prevention of mother-to-child transmission of HIV (PMTCT) program was not significantly affected by the lockdown, recorded by the PMTCT indicator. In this same study, significant declines in HIV testing were found during the lockdown, which also accounted for fewer overall health visits, through which routine HIV testing occurs. It was also shown that testing failed to return to the pre-COVID levels by December 2020 [22]. Employees of Anova Health, a District Support Partner for the US President’s Emergency Plan for AIDS Relief (PEPFAR), saw transitions from HIV testing services to COVID-19 testing services in five district provinces of South Africa until May 2020 [23].

Overall, the onset of the pandemic saw a reduction in access to community and in-person testing at most health facilities that also distribute HIV self-testing kits. Self-testing kits for HIV remained most conducive to the pandemic demands, as pick-up locations for the kits were offered for at-home use [35]. 

#### 4.1.3. Pre-Exposure Prophylaxis (PrEP)

The successful resolution of the HIV pandemic is contingent upon early detection measures such as screening and testing. Tenofovir-containing pre-exposure prophylaxis (PrEP) is currently recommended by the World Health Organization as a preventative measure for men and women with a high risk of contracting HIV. These individuals include women, young adults, men who have sex with men (MSM), and people who inject drugs (PWID), among others [36].

In Cape Town, a cohort study of HIV-negative pregnant and post-partum women in antenatal care revealed an increase of missed PrEP visits by 63% at the 1-month visit and 55% at the 3-month visit associated with the pandemic [24]. These measures were recorded at a primary care clinic with a high antenatal HIV prevalence that remained operational during the local lockdown. The implications of interrupted PrEP are of great significance to maternal and fetal health. A lack of adherence to regimen has shown to result in failure of pre-exposure prophylaxis of HIV [37]. Access to therapy to prevent HIV is essential in high-risk communities and must not be jeopardized by future lockdown events. Consequently, increasing emphasis on alternative PrEP delivery methods can prepare for prospective disturbances in care. 

### 4.2. Alternative Service Delivery

Even prior to the COVID-19 pandemic, many problems in the health system in South Africa faced challenges balancing the labor supply of HIV services, human resources, and the overall infrastructure of HIV healthcare services [38]. Patients were often discouraged from seeking care due to long waiting lines and disturbances to work schedules [39]. These issues warranted other methods of treatment reception, a need that was amplified after the pandemic hit. One study in South Africa found six different systems of alternative medication delivery, some put into place during the lockdown period. These options include outreach at the workplace, smart lockers, automated pharmacy dispensing units, alternative pickup points, adherence clubs, and home delivery, which was a popular option during the pandemic [25]. Much of this initiative was made possible through community healthcare workers and alternative transportation methods. At the same time, it is important to note the additional burden of instituting alternative delivery methods during lockdown among community healthcare workers, who make up a precarious labor force. 

Altogether, an increased accessibility of medication positively affects treatment adherence and overall outcomes for PLHIV. Home delivery is therefore a powerful tool in helping to achieve treatment goals and future prognoses. Through these means, health facilities were able to decongest without hindering the provision of care. Still, it is important to be wary that fees for delivery can create barriers to success, especially in more impoverished areas. 

### 4.3. Contraceptive Access

Decreases in the supply of condoms pose great risks to HIV susceptibility. Obstructions to access can greatly increase the risk of adverse outcomes in sexual and reproductive health, unplanned pregnancies, abortion, maternal mortality, and more [26]. Condoms are also critical to preventing the spread of other sexually transmitted diseases that may further reduce people’s immunological status. Contraceptive access was one of many routine healthcare services affected by the COVID-19 lockdown [27].

Promoting condom use was an existing issue in South Africa that was further challenged during the pandemic. For over 20 years, both male and female condoms have been used in the South African HIV prevention program, which employs condom distribution targets at public sector facilities [40,41]. One study revealed that many of these public sector sites had skewed distributions of condoms, increasing the unmet need for contraceptives in South Africa [26]. In this period, they found more limited access to condoms among the Black/African demographic and those in the Mpumalanga and KwaZulu-Natal Provinces, as well as those categorized in the third quintile of wealth. These findings highlight the need to increase community distribution of free condoms, especially in those regions that have higher rates of morbidity related to reproductive health. 

### 4.4. Symptomatic Admission

An interplay of the epidemiological and clinical features determines how diseases such as HIV are diagnosed. A single-site study in Cape Town found that the proportion of patients with HIV among those admitted for COVID-19 was higher than in the general population, and those living with HIV and hospitalized for COVID-19 have a higher mortality risk after adjusting for confounders, including older age, male sex, overweight/obesity, hypertension, diabetes, and active and previous tuberculosis [27,28]. The findings of this particular study opposed previous studies that dissociated HIV from a higher risk of COVID-19 illness severity and mortality.

### 4.5. Stigma

Health-related stigma continues to profoundly influence the prevention and care of HIV, with detrimental effects for those who experience stigma or avoid care due to a fear of stigma. In South Africa, a history of systemic racism and HIV-denying health policies have fomented a mistrust in government and health authorities, which may have resurfaced during the pandemic [34]. Medical disinformation about COVID-19 may have ultimately exacerbated this pre-existing mistrust in healthcare. Understanding the high risk of being stigmatized with COVID-19 among those living with HIV is critical, as the repercussions on health outcomes can be severe [29]. 

After the onset of the pandemic, the stigma of COVID-19 itself heightened other dubious conceptions about PLHIV. A prospective observational cohort study of PLHIV in South Africa reported over half of the participants either agreeing or strongly agreeing with survey questions assessing the stigma and stereotypes of COVID-19, as well as the lack of trust in healthcare providers [29]. It was determined that the association of stigma between HIV and COVID-19 did not occur during isolation. Rather, the stigma tapped into the deeply rooted history of systemic racism and the resulting impact on poverty and social determinants of health. These social justice issues have significantly contributed to the many health-related stigmas that PLHIV face. In the same study, several participants described the shame associated with revealing a positive COVID-19 status, confirming the internalized stigmas. Such sentiments can push PLHIV to be less open about initiating ART and ultimately lead to worse health outcomes [42]. To combat these possible events, an emphasis should be placed on the role of community healthcare workers and primary healthcare clinics. It is important to denote that, in clinics, HIV patients still face stigma and inadvertent disclosure of their status due to the behavior of clinic staff, jeopardizing the trust built with the community healthcare workers. At the intersection of care, community healthcare workers are one of the most important populations in bridging the gaps in trust due to stigmas against HIV and COVID-19 [43].

Overall, stigmas in healthcare can have compounding effects on patients, as they internalize the stigma and are discouraged from seeking out treatment options. The beginnings of the COVID-19 pandemic fomented mistrust while fueling stigmas, exposing already vulnerable populations to further discrimination. In these times, HIV care should remain transparent as we emphasize increased visibility and public health education to establish new norms that will break down stigmas. It will be necessary to listen and validate the concerns of this population to provide comfort and encouragement in seeking care. 

### 4.6. Ethical Modalities

There was no literature on the ethical implications qualified for inclusion in our systematic review due to the paucity of primary data. Still, it is important to consider the effects that COVID-19 had on the ethical considerations of HIV care. Arguments have been made on the importance of fairness and equity in COVID-19 care, emphasizing how the discrimination of patients with HIV can occur in ICU admissions. This lens investigates how the COVID-19 pandemic has increased stigma on existing vulnerable populations such as PLHIV, which directly affects the outcomes in patient care [44]. Thus, discrimination against PLHIV must be brought to the forefront to ensure equitable access to care during the COVID-19 pandemic and possible future outbreaks.

## 5. Strengths

To our knowledge, this review is the first of its kind to integrate the social, behavioral, and ethical aspects of COVID-19 on the HIV response in South Africa. This paper thoroughly reviews the primary literature on the impacts of COVID-19 lockdown on HIV services in all provinces of South Africa and explores these implications for future waves of the pandemic.

## 6. Limitations

Synthesizing an overhead analysis of the articles was limited by the availability of primary data on this topic. Many data sources did not have a study design that met the inclusion criteria for this review. Several model studies were published at the beginning of the pandemic, which predicted changes to the trajectory of HIV care in South Africa with the lockdown measures in place. These articles were ultimately not included, as the predicted results conflicted with the reality of the months that ensued after the initial lockdown. Furthermore, some papers that were found often had conflicting arguments due to the variations of the clinics across all provinces of South Africa. 

Many of the primary literature sources used in the data synthesis came from the South African District Health Information System, which had extensive data from the provinces but failed to provide context on the validity of the recorded numbers in each district. Additional limitations include challenges in the generalizability of the data. Differences in locations, populations, and service structures made it challenging to account for interstudy variability when the findings between studies were different. Areas of HIV prevention and service that were addressed by only one manuscript may not be generalizable to the entire population. As a result of delays in data gathering, analysis, peer review, and publication, most of the manuscripts reflected data from the first months of the COVID pandemic and thus may not reflect the current status of the services that were affected in the early months. For instance, one study found that people living with HIV were overrepresented in COVID-19 hospitalizations in Cape Town during the first 6 weeks of the pandemic, but it is unclear if this trend persisted throughout the pandemic.

## 7. Conclusions

It is evident that South Africa has taken great responsibility in the management of HIV/AIDS within recent years. However, the advent of COVID-19 has highlighted continued areas for improved delivery, outcomes, and equity in care with the concurrent epidemic of HIV. Our 15 articles revealed how paucities in communication, disturbances in the medical supply chain, alternative delivery of care affecting cost and medication disbursement, barriers to contraceptive access, and long-established stigmas of HIV came into the public eye as effects of the COVID-19 pandemic. Moving forward, the mainstay of the focus should be to reduce the burden on individuals in society through policymaking that addresses both infectious diseases. Just as South Africa has led the way in instituting innovative approaches to HIV care to overcome past challenges, it must again proactively prevent future delays and interruptions in healthcare services for people living with HIV in the context of the COVID pandemic. We must continue to investigate the history of how we respond to public health emergencies, noting the benefits and repercussions of each measure taken to progress the science of the disease while managing the healthcare services. Furthermore, preparation strategies should be implemented with a focus on equitable care. 

To our knowledge, this is the first review of its kind to gather comprehensive data on the social, behavioral, and ethical effects of the COVID-19 pandemic on HIV care in South Africa. We propose this review as a learning tool for more robust measures to prevent disruptions in HIV care, reduce the stigmatization of HIV, promote social connectedness, and push the need for alternative methods of delivery services to patients and high-risk populations.

## 8. Protocol Registration

This systematic review is registered in the PROSPERO database of systematic reviews.

## Figures and Tables

**Figure 1 ijerph-19-09766-f001:**
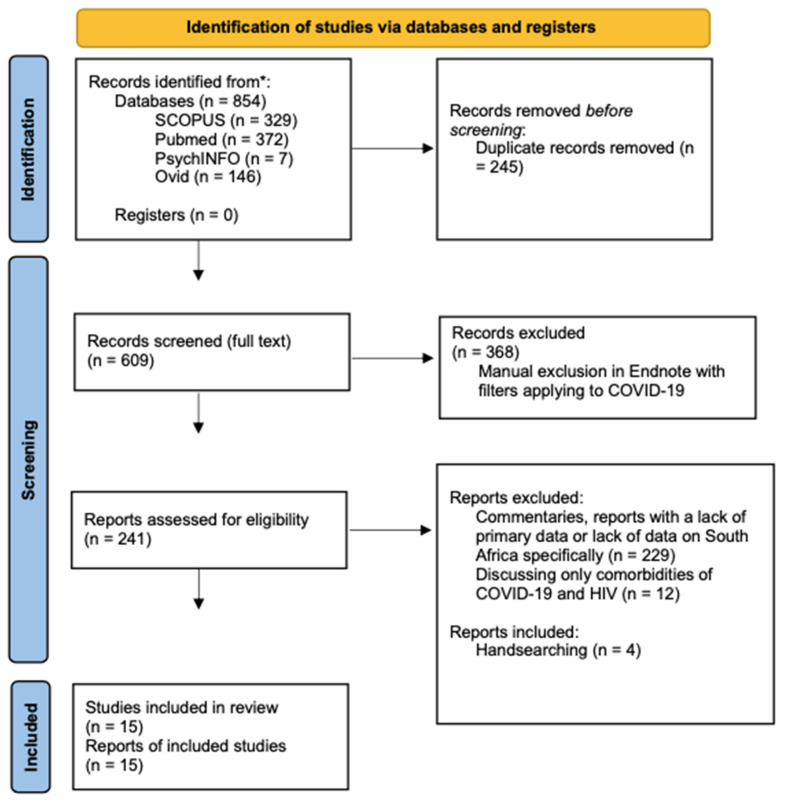
PRISMA flow diagram. * Developmental process for the Preferred Reporting Items for Systematic Reviews and Meta-analyses (PRISMA) checklist. This figure demonstrates the search strategy and database search for cohort studies, qualitative studies and systematic reviews included in the literature analysis.

**Figure 2 ijerph-19-09766-f002:**
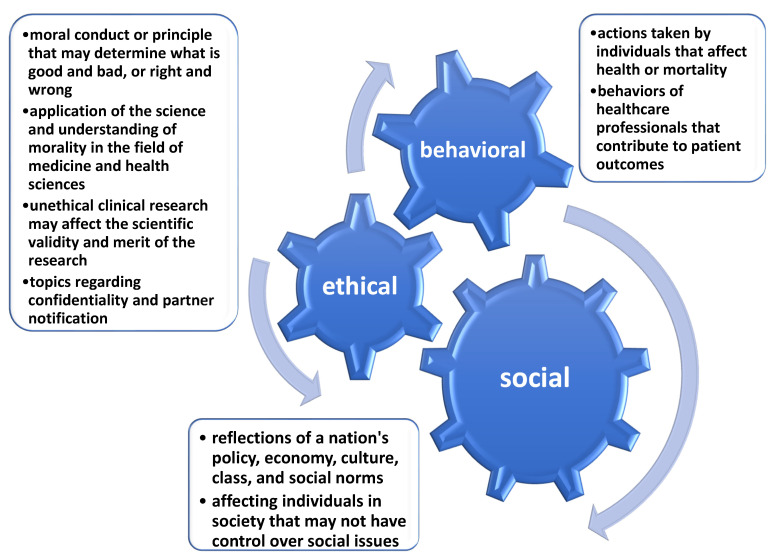
Defining the social, behavioral, and ethical lenses. Note: See [8,12,13,14,15,16].

**Table 1 ijerph-19-09766-t001:** Cohort Study Checklist.

	Section A: Are the Results of the Study Valid?						
Author	Did the Study Address a Clearly Focused Issue?	Was the Cohort Recruited in an Acceptable Way?	Was the Exposure Accurately Measured to Minimize Bias?	Was the Outcome Accurately Measured to Minimize Bias?	5. (a) Have the Authors Identified all Important Confounding Factors?	5. (b) Have they Taken Account of the Confounding Factors in the Design and/or Analysis?	6. (a) Was the Follow Up of Subjects Complete Enough?
Dorward	yes	yes	yes	yes	no	no	no
Siedner	yes	yes	yes	yes	yes	yes	yes
Benade	yes	yes	yes	yes	yes	no	NA
Western Cape Department of Health in collaboration with the National Institute for Communicable Diseases, South Africa	yes	yes	yes	yes	yes	yes	NA
El-Krab	yes	yes	yes	yes	yes	yes	Cannot tell
Jensen	yes	yes	yes	yes	yes	no	Cannot tell
Mutyambizi	yes	yes	yes	yes	yes	no	NA
Rees	yes	yes	yes	yes	yes	yes	Cannot tell
Davey	yes	yes	yes	yes	no	no	NA
Boralinwa	yes	yes	yes	yes	yes	yes	NA
Pillay	yes	yes	yes	yes	yes	yes	NA
Parker	yes	yes	yes	yes	yes	yes	NA
Jarolimova	yes	yes	yes	yes	yes	yes	no
	** Section B: What are the Results? **				** Section C: Will the Results Help Locally? **		
	What are the Results of the Study?		How Precise are the Results?	Do you Believe the Results?	Will the Results Help Locally?	Do the Results of this Study Fit with Other Available Evidence? All Important Confounding Factors?	What are the Implications of this Study for Practice?
Dorward	ART treatments were generally maintained during 2020 lockdown, but HIV testing and ART initiation was impacted.		Yes, CI intervals and significance levels were reported.	yes, the statistical analysis holds.	yes	Yes and no, other studies show some drop-off in treatment continuance and maintenance, whereas this study shows a steady maintenance of ART tx. Could depend on specific pop examined.	Meaning that during a community lockdown, strategies need to be implemented to maintain or increase testing and initiation of treatments to further prevention efforts.
Siedner	No changes were found in total clinic visits/clinic/day at the time of starting the level 5 lockdown.		Statistical significance and CI were stated	yes	yes	Does not fit with other studies in the same province that do suggest reductions in visit during this time.	This would implicate a lesser impact of the COVID-19 pandemic on HIV care in the KZN Province.
Benade	Initiations of ART were 20% fewer in 2020 compared to 2019, with large declines in all provinces between April and June 2020. These numbers remained low for the remainder of 2020, other than short periods of recovery between COVID-19 waves and possible improvement starting in March 2021		CI not reported	Yes, to some degree. Authors mentioned a lack of validation of the data from the DHIS database.	yes	yes	Lack of data validation may reduce support for the paper’s argument
Western Cape Department of Health in collaboration with the National Institute for Communicable Diseases, South Africa	Among 601 hospitalized PLWH, 33% had their CD4 count measured during the COVID-19 episode, of which 35% had CD4 count <200 cells/µL, which was associated with COVID-19 death. HIV and previous tuberculosis infection was associated with death but not current TB.		HRs and CIs are reported	Yes, the statistical analysis holds.	yes	yes	Provides context on TB/HIV coinfection and comorbidities for HIV
El-Krab	Food access was limited among the majority of the sample. Over half of the sample could not work or attend school due to lockdown policies. More than one in three participants revealed interruptions in their HIV care, specifically in medication access.		Yes, chi-square tests and *t*-tests were used to assess correlations, with statistical significance noted.	yes, the statistical analysis holds.	yes	Yes, authors discussed similar results with other primary literature sources	Supports focus on psychological well-being with adherence of ART associated with COVID-19 protections
Jensen	Reductions in hospital visits were recorded with less recovery than PHC clinic attendance. Access to service was reduced further for young children than for adults and adolescents.		Yes, CI intervals and significance levels were reported.	yes, the statistical analysis holds.	yes	Cannot tell	Confirmation of the anticipated impact of the COVID-19 pandemic on child health services, with a pattern of disruption across multiple data elements and indicators covering service access, service delivery and child wellbeing.
Mutyambizi	Study showed statistically significant declines in ART initiation, both at the beginning of the first wave in April and at the move to lockdown level 1 in September.		single-group ITSA regression model was used	yes, the statistical analysis holds.	yes	Yes and no, authors mentioned finding data similar to another study covering DHIS data in all of SA but in contrast to a DHIS data set in the KZN Province	Comprehensive data analysis that can help prepare for breaks in treatment in future waves
Rees	Rates of COVID-19 infection were as high in frontline support staff as those in clinical HCWs.		Cumulative incidence was noted, but CI or p values were not stated	somewhat, could use some more statistical data validation	yes	Cannot tell	CHW of HIV facilities who test positive for COVID-19 pose infectious risk to PLHIV, potentially affecting the prognosis
Davey	During the initial lockdown, women’s missed PrEP visits increased from 34% (pre-lockdown) to 57% (during lockdown).		Yes, IQR intervals and CI levels reported	yes, the statistical analysis holds.	yes	Cannot tell	Clinical implications for missed visits are great for maternal and infant health. Pregnant and post-partum women who were on PrEP cited their reasoning for missed appts. to fear of contracting the virus. Making it essential for different delivery of care in the community to address these barriers and continue prevention efforts.
Boralinwa	One-fourth of South Africans couldn’t access condoms during the pandemic; individuals in lower wealth groups had less public access to condoms; ppl w/lower educational attainment, between 25 and 34 were less likely to prefer public source of condoms		CI levels, agency levels, and sig. levels reported	yes, the statistical analysis holds.	yes	yes	Study shines light on access to condoms that was limited during the pandemic and the preferred method of obtaining condoms, which was skewed across the public and age cohorts. Strategies on community distribution of free condoms should be implemented in the future.
Pillay	Confirmed decrease in use of primary healthcare facilities within all South African provinces		CI or p values were not stated	somewhat, could use some more statistical data validation	yes	Yes, described similar findings from a review by the Global Fund	
Parker	PLHIV with COVID-19 may have a high probability of admission to hospital, but had similar presentations, comorbidities and outcomes when compared with the HIV-negative study population.		Statistical significance and CI were stated	Mostly, although the sample size was small	yes	Cannot tell	The presentation and outcome of patients with HIV did not differ significantly from those of patients without HIV
Jarolimova	High stigma of COVID-19 was associated more with the female gender and previous HIV stigma. Lower stigma of COVID-19 was associated with television broadcast as an information source. Further efforts should focus on stigma and mistrust, as well as its effects on protective health behaviors and vaccine hesitancy.		Descriptive statistics with univariate logistic regression models	yes, the statistical analysis holds.	yes	Cannot tell	Sheds light on the prevalence of medical mistrust and conspiracy beliefs related to COVID-19 among PLWH in South Africa

**Table 2 ijerph-19-09766-t002:** Systematic Review Checklist.

Author	Section A: Are the Results of the Review Valid?					
	1. Did the Review Address a Clearly Focused Question?	2. Did the Authors Look for the Right Type of Papers?	3. Do You Think all the Important, Relevant Studies were Included?	4. Did the Review’s Authors do Enough to Assess Quality of the Included Studies?	5. If the Results of the Review have been Combined, was it Reasonable to do so?	6. Apart from the Experimental Intervention, did Each Study Group Receive the Same Level of Care (that is, Were They Treated Equally)?
Mash	yes	yes	yes	yes	yes	NA
** Section B: What are the Results? **				** Section C: Will the Results Help Locally? **		
	**6. What are the Overall Results of the Review?**		**7. How Precise are the Results?**	**8. Can the Results be Applied to the Local Population?**	**9. Were all Important Outcomes Considered?**	**10. Are the Benefits Worth the Harms and Costs?**
	The study suggests the implementation of a hybrid system that allows alternatives to heed to the needs of each patient.		Precise, studies were adequately analyzed and reviewed	Yes, promotes the hybrid approach in South Africa	yes	yes

**Table 3 ijerph-19-09766-t003:** Qualitative Study Checklist.

Author	Section A: Are the Results of the Review Valid?						
	1. Was There a Clear Statement of the Aims of the Research?	2. Is a Qualitative Methodology Appropriate?	3. Was the Research Design Appropriate to Address the Aims of the Research?	4. Was the Recruitment Strategy Appropriate to the aims of the Research?	5. Was the Data Collected in a Way that Addressed the Research Issue?	6. Has the Relationship between Researcher and Participants been Adequately Considered?	7. Have Ethical Issues been Taken into Consideration?
Grimsrud	yes	yes	yes	yes	Yes	yes	yes
	** Section B: What are the Results? **				** Section C: Will the Results Help Locally? **		
	**7. Have Ethical Issues been Taken into Consideration?**		**8. Was the Data Analysis Sufficiently Rigorous?**		**9. Is There a Clear Statement of Findings?**		**10. How Valuable is the Research?**
	yes		yes		yes		The authors review and promote the use of differentiated service delivery for HIV, which extends access to treatment services

CASP quality assessment tools can be viewed in Excel format here: https://tinyurl.com/mutmuzx2 (accessed on 22 June 2022).

**Table 4 ijerph-19-09766-t004:** Breakdown of the included studies.

Ref	Author	Study Type	Study Population	N	Analytical Method
[1]	Dorward	Interrupted time series analysis	People testing for HIV, initiating ART, and collecting ART at participating clinics recorded on the DHIS	3,706,543	multivariable
[4]	Siedner	Interrupted time series analysis	Patients from 11 primary healthcare clinic in KwaZulu-Natal Province with data recorded on the Africa Health Research Institute (AHRI) surveillance system	46,523	multivariable
[17]	Benade	Retrospective cohort	Facilities providing ART initiations in SA District Health Information System (DHIS)	2471	multivariable
[18]	Western Cape Department of Health	Population cohort	Adults attending public sector health facilities in Western Cape	3,460,932	Univariable and multivariable HRs
[19]	Grimsrud	Qualitative study	ART receiving patients in sub-Saharan Africa	NA	univariable
[20]	El-Krab	Observational cohort	Patients receiving services for HIV treatment at a public health clinic in an established formal township of Cape Town	272	univariable
[21]	Jensen	Retrospective cohort	Health facilities recorded in the DHIS data set for KwaZulu-Natal Province	681	multivariable
[22]	Mutyambizi	Retrospective cohort	Health facilities of the Mopani District in the Limpopo Province registered on DHIS	NA	multivariable
[23]	Rees	Retrospective cohort	Anova Health Institute employees of primary healthcare facilities in Cape Town, Capricorn, Sedibeng, Johannesburg and Mopani districts in South Africa	562	univariable
[24]	Davey	Population cohort	Pregnant and post-partum women with HIV-negative status in antenatal care at a primary care clinic that was operational during the COVID-19 lockdown in a Cape Town community with high antenatal HIV prevalence	455	univariable
[25]	Mash	Systematic review	Reviews of alternative mechanisms for delivery of medication to South African primary health clinic patients	4253	univariable
[26]	Bolarinwa	Observational cohort	Respondents of National Income Dynamics Study-COVID Rapid Mobile Survey	5304	univariable
[27]	Pillay	Observational cohort	Patients making visits for primary healthcare, reproductive, maternity, and HIV care within all African provinces recorded on DHIS	NA	multivariable
[28]	Parker	Retrospective cohort	Patients admitted to the Tyberg hospital in Cape Town	116	multivariable
[29]	Jarolimova	Observational cohort	Patients receiving HIV care in 9 DOH primary health clinics that enroll in the Central Chronic Medicines Dispensing and Distribution program (CCMDD) in the urban Umlazi township	303	multivariable

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
