# Peer review of "The Social, Behavioral, and Ethical Modalities of COVID-19 on HIV Care in South Africa: A Systematic Review"

_ijerph, 2022, doi:10.3390/ijerph19159766_

Round 1
Reviewer 1 Report
Thank you for the opportunity to review your manuscript. It was well written and covers an important topic.
I have the following concerns that should be addressed.
Line 23 (abstract), Because ethics is an essential part of the manuscript and review, I think you should include a rationale (not data-based) for why these articles (n = 2) were excluded in the abstract.
Line 77, I believe the abbreviation "ATR" should be "ART."
Line 97, the phrase "field of medicine and health sciences" ought to be in the plural form "fields of medicine and health sciences."
Line 99, Consider whether you should expand to healthcare providers. Physicians are not the only health care providers who have helped South Africa achieve its progress toward the 90-90-90 objectives. For example, South African nurses are providing exceptional leadership in improving ART adherence and HIV prevention and care through Nurse Initiated Antiretroviral Therapy (NIMART) programs. Without naming the other health care professionals who contribute to this work, you discount the work of those professionals. Additionally, you run the risk of implying that those professionals have mastered care provision and provide ethical care. For effective and successful NIMART programs, nurses must also provide ethical care. If the literature you reviewed only revealed research into the work of physicians you should speculate about why that occurred and include in your discussion what challenges from social, ethical, and behavioral perspectives this creates by excluding other health care professionals from addressing the public health crises of COVID-19 and HIV and what implications those exclusions would have for responding to future pandemics. This is especially problematic given the number of nurses who are essential to a well-functioning healthcare system and public health response.
Line 105, Building from my previous comment at line 99, if your objective is to equip South Africa for future pandemics, it is essential to include all of the health disciplines that contribute to a well-functioning public health system - not just physicians.
Line 182 (Figure 2), In the behavioral domain, consider expanding "physicians" to "health care providers" because physicians are not the only healthcare professionals working in the healthcare system and they are not present in sufficient numbers to address the magnitude of the HIV and COVID-19 pandemics.
Line 182 (Figure 2), in the ethical domain, point 2 should be "fields of medicine and health sciences."
Line 313, Be consistent in the use of "among" versus "amongst" throughout the manuscript.
Author Response
- Line 23 (abstract), Because ethics is an essential part of the manuscript and review, I think you should include a rationale (not data-based) for why these articles (n = 2) were excluded in the abstract.
- Thank you very much for the suggestion. We have updated the abstract section to include this.
- Line 77, I believe the abbreviation "ATR" should be "ART."
- This edit has been made.
- Line 97, the phrase "field of medicine and health sciences" ought to be in the plural form "fields of medicine and health sciences."
- This edit has been made.
- Line 99, Consider whether you should expand to healthcare providers. Physicians are not the only health care providers who have helped South Africa achieve its progress toward the 90-90-90 objectives. For example, South African nurses are providing exceptional leadership in improving ART adherence and HIV prevention and care through Nurse Initiated Antiretroviral Therapy (NIMART) programs. Without naming the other health care professionals who contribute to this work, you discount the work of those professionals. Additionally, you run the risk of implying that those professionals have mastered care provision and provide ethical care. For effective and successful NIMART programs, nurses must also provide ethical care. If the literature you reviewed only revealed research into the work of physicians you should speculate about why that occurred and include in your discussion what challenges from social, ethical, and behavioral perspectives this creates by excluding other health care professionals from addressing the public health crises of COVID-19 and HIV and what implications those exclusions would have for responding to future pandemics. This is especially problematic given the number of nurses who are essential to a well-functioning healthcare system and public health response.
- Thank you very much for bringing this to our attention. It is very important to acknowledge the healthcare team and give credit where it is due. The implications of excluding non-physician providers are great. This has been corrected.
- Line 105, Building from my previous comment at line 99, if your objective is to equip South Africa for future pandemics, it is essential to include all of the health disciplines that contribute to a well-functioning public health system - not just physicians.
- This has been corrected.
- Line 182 (Figure 2), In the behavioral domain, consider expanding "physicians" to "health care providers" because physicians are not the only healthcare professionals working in the healthcare system and they are not present in sufficient numbers to address the magnitude of the HIV and COVID-19 pandemics.
- This portion of the graphic has been revised.
- Line 182 (Figure 2), in the ethical domain, point 2 should be "fields of medicine and health sciences."
- This portion of the graphic has been revised.
- Line 313, Be consistent in the use of "among" versus "amongst" throughout the manuscript.
- Thanks for the comment. This consistency has been established.
Reviewer 2 Report
I appreciate the authors efforts in including social, behavioral, and ethical factors to evaluate the impact of the pandemic on the HIV epidemic. However, I have a few concerns listed below:
1. I understand that each manuscript was assessed to determine quality, validity, and biases; however, for the topics that are covered by more than one manuscript, how are inter-study variabilities controlled for? For instance if rates are reported, are they in agreement?
2. For included topic areas that are referenced by only one manuscript, do the authors feel that this is substantial and accurate to generalize it to the entire population?
3. For rates that declined during the pandemic but are rebounding, are they near pre-COVID levels? In particular, a significant drop in clinic visits was reported during the lockdown and followed with a 20% increase. Does the bring the visits back to baseline? If so, wouldn't that mean that the usual supply should be able to manage demand. Or is this a 20% increase above the pre-COVID visits? Also, did the decrease in visits result in supply stock that can address the current demand?
4. There was a brief mention (section 4.5) of the higher proportion of HIV+ individuals in those admitted for COVID-19 compared to the general population. This is probably one of the most interesting reports; however, this section is very brief and vague. With COVID being around indefinitely, it would be interesting to see how vulnerable immunocompromised individuals are affected. How does this co-infection affect some of the areas that are discussed?
Author Response
Q 1. I understand that each manuscript was assessed to determine quality, validity, and biases; however, for the topics that are covered by more than one manuscript, how are inter-study variabilities controlled for? For instance if rates are reported, are they in agreement?
- A1. Thank you for the question. Indeed, each included title was assessed by two of the study team for quality assurance. To deal with inter-study variability, the team adopted the notion of reporting and discussing these variabilities with proper citation. Whenever possible, other articles (outside the included titles) are to be consulted and cited to support one value or further explain the context of the variations.
Q2. For included topic areas that are referenced by only one manuscript, do the authors feel that this is substantial and accurate to generalize it to the entire population?
- A2. Thank you very much for the question. As a systematic review, the authors assumed the highest accuracy in reporting the findings of different included studies and properly discussing them. In this sense, the findings of different studies were reported and discussed within the limitations of their methods and study samples. Authors have never claimed generalization of specific findings without being assured that the methods of the studies generated these findings are pertinent to generalization.
Q3. For rates that declined during the pandemic but are rebounding, are they near pre-COVID levels? In particular, a significant drop in clinic visits was reported during the lockdown and followed with a 20% increase. Does the bring the visits back to baseline? If so, wouldn't that mean that the usual supply should be able to manage demand. Or is this a 20% increase above the pre-COVID visits? Also, did the decrease in visits result in supply stock that can address the current demand?
- A3. Thanks for the comment. The 20% increase was over the baseline. The study referred the higher utilization to the increase in demand due to the interruption of services during the lockdown. We have addressed this comment (Please refer to lines 213 and 214). In the province of KwaZulu-Natal, a significant drop in clinic visits during lockdown was immediately followed by a 20% increase in visits for HIV over the original baseline. The higher utilization may reflect the increase in demand due to the interruptions of services during the lockdown [41].
Q4. There was a brief mention (section 4.5) of the higher proportion of HIV+ individuals in those admitted for COVID-19 compared to the general population. This is probably one of the most interesting reports; however, this section is very brief and vague. With COVID being around indefinitely, it would be interesting to see how vulnerable immunocompromised individuals are affected. How does this co-infection affect some of the areas that are discussed?
- A4. Thanks for the comment. More explanation was added. Please refer to section 4.5.
- An interplay of epidemiological and clinical features determine how diseases like HIV are diagnosed. A single site study in Cape Town found that the proportion of patients with HIV among those admitted for COVID-19 was higher than in the general population and those living with HIV and hospitalized for COVID-19 have a higher mortality risk after adjusting for confounders, including older age, male sex, overweight/obesity, hypertension, diabetes, active and previous tuberculosis [35]. The findings of this particular study opposed previous studies that dissociated HIV from a higher risk of COVID-19 illness severity and mortality.
Thank you very much for the suggestions. We have expanded on some of theses concerns further by adding to the limitations section.
- Limitations were expanded on to Additional limitations include challenges in the generalizability of data. Differences in location, population, and service structures made it challenging to account for inter-study variability when findings between studies were different. Areas of HIV prevention and service that were addressed by only one manuscript may not be generalizable to the entire population. As a result of delays in data gathering, analysis, peer review, and publication, most manuscripts reflect data from the first months of the COVID pandemic, and thus may not reflect the current status of services that were affected in the early months. For instance, one study found that people living with HIV were overrepresented in COVID-19 hospitalizations in Cape Town during the first 6 weeks of the pandemic, but it is unclear if this trend persisted throughout the pandemic.
Reviewer 3 Report
Thank you for inviting me to review this manuscript. The paper is well written and thorough systematic review. The topic of increased HIV cases and reduced care throughout the pandemic is a cause for concern, so this paper would add to the research.
I have a few concerns:
1. 1.0- 1.3 is very well written except for this item on Page 2, line 87. Evinced? Do you mean ‘evidenced’?
2. Page 3 line138. The sentence begins with 4. Should it be Four per APA?
3. Page 4, lines 151-154. CASP checklist is a questionable tool (used mainly in workshops) for use in this paper. Please expand on how it was helpful in establishing the use of cohort studies, systematic reviews, and qualitative studies in the literature search.
4. Page 17. Multiple sentences begin with a number rather than a word. Is this acceptable formatting?
5. The paper specifies 15 studies chosen, yet only 14 were included in the review of the articles on page 17, lines 169-180.
6. The main goal of the paper was to address the impact of Covid 19 pandemic on the HIV epidemic and present an argument on better equipping the country for future public health outbreaks. With this said, the abstract and conclusion left me searching for more information. While you discuss the future direction of HIV care during a pandemic, highlighting what you found impacted the HIV epidemic during the pandemic (in a sentence or two) may help the reader both in the abstract and conclusion. Stating that your 15 studies identified transparency of care, alternative delivery of care affecting cost and medication disbursement, contraceptive access, etc., would help.
Author Response
- 1.0- 1.3 is very well written except for this item on Page 2, line 87. Evinced? Do you mean ‘evidenced’?
- Thank you very much. Yes, this has been revised.
- Page 3 line138. The sentence begins with 4. Should it be Four per APA?
- Thank you for bringing this to our attention. Yes, this has been revised.
- Page 4, lines 151-154. CASP checklist is a questionable tool (used mainly in workshops) for use in this paper. Please expand on how it was helpful in establishing the use of cohort studies, systematic reviews, and qualitative studies in the literature search.
- Thanks for the comment. Prior to the CASP tool, we attempted to use a couple other qualitative tools to measure the validity of the chosen studies. Some members of the team advised against using these, as they were based on point by point scales and/or letter grades to assess the studies. It was suggested that the authors of these articles may find this tactic harsh or offensive. Ultimately, we sought the CASP tool which was comprehensive to all study designs and implemented quality in the judgement-making process. Although it is commonly used in workshops, we believe that it prompted many essential questions to assess the research, study design, bias, etc.
- Page 17. Multiple sentences begin with a number rather than a word. Is this acceptable formatting?
- Thank you for bringing this to our attention. This paragraph has been revised.
- The paper specifies 15 studies chosen, yet only 14 were included in the review of the articles on page 17, lines 169-180.
- This section has been revised.
- The main goal of the paper was to address the impact of Covid 19 pandemic on the HIV epidemic and present an argument on better equipping the country for future public health outbreaks. With this said, the abstract and conclusion left me searching for more information. While you discuss the future direction of HIV care during a pandemic, highlighting what you found impacted the HIV epidemic during the pandemic (in a sentence or two) may help the reader both in the abstract and conclusion. Stating that your 15 studies identified transparency of care, alternative delivery of care affecting cost and medication disbursement, contraceptive access, etc., would help.
- Thank you very much for this suggestion. This has been added to the conclusion section.
- Line 432
- It is evident that South Africa has taken great responsibility in the management of HIV/AIDS within recent years. However, the advent of COVID-19 has highlighted continued areas for improved delivery, outcomes, and equity in care with the concurrent epidemic of HIV. Our 15 articles revealed how paucities in communication, disturbances in medical supply chain, alternative delivery of care affecting cost and medication disbursement, barriers to contraceptive access and long-established stigmas of HIV came into the public eye as effects of the COVID-19 pandemic. Moving forward, the mainstay of focus should be to reduce the burden on individuals in society through policymaking that addresses both infectious diseases. Just as South Africa has led the way in instituting innovative approaches to HIV care to overcome past challenges, it must again proactively prevent future delays and interruptions in healthcare services for people living with HIV in the context of the COVID pandemic. We must continue to investigate the history of how we respond to public health emergencies, noting the benefits and repercussions of each measure taken to progress the science of disease while managing healthcare services. Furthermore, preparation strategies should be implemented with a focus on equitable care.
- Line 432
- Thank you very much for this suggestion. This has been added to the conclusion section.